# *Porphyromonas gingivalis* HmuY and *Bacteroides vulgatus* Bvu—A Novel Competitive Heme Acquisition Strategy

**DOI:** 10.3390/ijms22052237

**Published:** 2021-02-24

**Authors:** Klaudia Siemińska, Patryk Cierpisz, Michał Śmiga, Teresa Olczak

**Affiliations:** Laboratory of Medical Biology, Faculty of Biotechnology, University of Wrocław, 14A F. Joliot-Curie St., 50-383 Wrocław, Poland; klaudia.sieminska@uwr.edu.pl (K.S.); patryk.cierpisz2@uwr.edu.pl (P.C.); michal.smiga2@uwr.edu.pl (M.Ś.)

**Keywords:** *Porphyromonas gingivalis*, *Bacteroides vulgatus*, HmuY, microbiome, gut, heme, hemophore, iron

## Abstract

Human oral and gut microbiomes are crucial for maintenance of homeostasis in the human body. *Porphyromonas gingivalis*, the key etiologic agent of chronic periodontitis, can cause dysbiosis in the mouth and gut, which results in local and systemic infectious inflammatory diseases. Our previous work resulted in extensive biochemical and functional characterization of one of the major *P. gingivalis* heme acquisition systems (Hmu), with the leading role played by the HmuY hemophore-like protein. We continued our studies on the homologous heme acquisition protein (Bvu) expressed by *Bacteroides vulgatus,* the dominant species of the gut microbiome. Results from spectrophotometric experiments showed that Bvu binds heme preferentially under reducing conditions using Met145 and Met172 as heme iron-coordinating ligands. Bvu captures heme bound to human serum albumin and only under reducing conditions. Importantly, HmuY is able to sequester heme complexed to Bvu. This is the first study demonstrating that *B. vulgatus* expresses a heme-binding hemophore-like protein, thus increasing the number of members of a novel HmuY-like family. Data gained in this study confirm the importance of HmuY in the context of *P. gingivalis* survival in regard to its ability to cause dysbiosis also in the gut microbiome.

## 1. Introduction

Human oral and gut microbiomes are crucial for the maintenance of homeostasis in the human body. A healthy gut microbiome is colonized by bacteria belonging mostly to the phylum Bacteroidetes, with *Bacteroides vulgatus* being the dominant species [1,2,3]. *B. vulgatus* exhibits immunomodulating properties important, for example, in preventing the induction of colitis [4]. In patients suffering from coronary artery disease, lower abundance of *B. vulgatus* has been observed, and oral gavage with live bacteria was shown to protect mice against the disease [5], suggesting that the presence of this bacterium might help prevent development of atherosclerosis. However, altered or disturbed mutualism between gut microbiome members results in dysbiosis with the subsequent development of intestinal and systemic diseases such as inflammatory bowel disease, intra-abdominal sepsis, endocarditis, and atherosclerosis [5,6]. In the oral microbiome, dysbiosis results in periodontal diseases, which influence the onset and progression of systemic diseases such as diabetes, atherosclerosis, and Alzheimer’s disease [7,8,9,10,11]. Importantly, periodontopathogens, not only those residing in the oral cavity but also those invading the gut, are able to influence host health and participate in systemic diseases [12,13,14,15,16,17]. One of the best examples is *Porphyromonas gingivalis*, the key etiologic agent of chronic periodontitis, which can invade the gut and cause dysbiosis, and therefore can be involved in local and systemic inflammatory diseases.

The high bacterial density in the oral cavity and gut causes intense competition to acquire nutrients among microbiome residents. For members of the phylum Bacteroidetes, iron and heme (the term heme is used in this study regardless of iron valence) are regarded as essential nutrients and key virulence factors [18,19,20,21,22]. *P. gingivalis* does not produce and use siderophores to gain iron, lacks the ability to synthesize protoporphyrin IX (PPIX), and therefore must utilize mechanisms to acquire these compounds. Among them is the best characterized Hmu system composed of a hemophore-like, heme-binding protein (HmuY), TonB-dependent outer-membrane receptor involved in heme transport through the outer membrane (HmuR), and four uncharacterized proteins with unknown functions [19]. Other *P. gingivalis* mechanisms of iron and heme uptake comprise Hus system and a set of cysteine proteases, gingipains [19].

We identified the presence of and characterized HmuY homologs also in other periodontopathogens, namely in *Tannerella forsythia* (Tfo) and *Prevotella intermedia* (PinO and PinA) [23,24]. We also found that periodontal HmuY homologs [23,24] and *Streptococcus gordonii* glyceraldehyde-3-phosphate dehydrogenase (GAPDH) [25] may bind heme and subsequently provide *P. gingivalis* with this essential nutrient. To date, iron and heme acquisition systems in *B. vulgatus*, a dominant member of the healthy gut microbiome, have not been characterized. Our main hypothesis is that the HmuY-based heme acquisition mechanism expressed by Bacteroidetes can be used to fulfill not only their nutritional requirements but also to increase their virulence. Importantly, competition in heme acquisition may occur not only between *P. gingivalis* and cohabitating periodontopathogens, but also between *P. gingivalis* and gut bacteria, thus providing *P. gingivalis* with heme and enabling it to play a keystone pathogen role in both microbiomes.

To obtain a desired breakthrough in the field of understanding and treatment of infectious inflammatory diseases, it is first necessary to characterize the proteins crucial to the growth and virulence of key members of human microbiomes responsible for the dysbiosis. Therefore, we aimed to continue our studies on the heme acquisition mechanism also in the gut microbiome with the main role played by *P. gingivalis* HmuY and its homolog expressed by *B. vulgatus*.

## 2. Results and Discussion

### 2.1. B. vulgatus Bvu May Belong to the HmuY-Like Family of Hemophore-Like Proteins

Our previous work resulted in extensive biochemical and functional characterization of one of the major *P. gingivalis* heme acquisition systems (Hmu) [19]. Genetic organization of respective genes located in the *P. gingivalis hmu* operon and in the *hmu*-like operons in other Bacteroidetes members so far characterized [23,24] are different compared to their counterparts, typical *hem* operons found in other Gram-negative bacteria [26,27]. *B. vulgatus* ATCC 8482, similar to *P. gingivalis*, possesses one HmuY homolog (locus ID: BVU_2192 or BVU_RS11035) (Figure 1), termed here Bvu. Although the three-dimensional structure of the Bvu protein in apo form was solved (PDB ID: 3U22), the protein was not biochemically and functionally characterized. Both the amino acid alignment (Figure 2) and comparison of three-dimensional apo-protein structures (Figure 3) demonstrated that Bvu may belong to the family of HmuY-like heme-binding proteins. Therefore, we aimed to examine heme-binding properties of Bvu protein and its interaction with hemoproteins.

### 2.2. B. vulgatus Bvu Is Expressed at Higher Levels under Low-Iron Conditions

Previously, we demonstrated that *P. gingivalis* HmuY is constitutively expressed at low levels. However, when the bacterium grew in vitro under low-iron/heme conditions or as a biofilm constituent, creating an environment typical for the healthy oral cavity or for early stages of chronic periodontitis, significantly higher levels of HmuY at both mRNA and protein levels can be observed [28,29]. We also demonstrated that HmuY is important for bacterial survival and the invasion of human cells [20]. Significantly higher HmuY expression in *P. gingivalis* grown in the form of biofilm together with *Candida albicans* [30] and in patients with periodontitis [31,32] confirmed its requirement under in vivo conditions. Recently, we also showed that *T. forsythia* and *P. intermedia* produce higher levels of their HmuY homologs when grown under low-iron/heme conditions [23,24]. In this study, we showed that, analogously to the so far characterized members of the HmuY-like family of proteins, Bvu was also produced at higher levels when bacteria were cultured in a medium limited in iron and without added heme. Quantitative reverse transcriptase-polymerase chain reaction (qRT-PCR) analysis demonstrated a higher transcript level in starved bacteria (1440 ± 533-fold change) as compared to bacteria grown in rich culture medium. This suggests that Bvu could be engaged in heme acquisition as a sole iron source.

### 2.3. B. vulgatus Bvu Binds Heme in A Manner Different to P. gingivalis HmuY but Similar to Other HmuY Homologs

Detailed characterization of the HmuY-heme complex has demonstrated that both Fe(III)heme and Fe(II)heme are in a low-spin hexa-coordinate environment in the protein and that His134 and His166 are heme iron-coordinating ligands [33,34]. The purified Bvu protein in a complex with heme under air (oxidizing) conditions showed a UV-visible spectrum with the broad Soret band maximum at 388–401 nm, additional shoulders at 499 and 532 nm, and the charge transfer CT1 band maximum at 608–611 nm (Figure 4A and Figure 5). These features may suggest the presence of penta-coordinate high-spin ferric heme protein. The sixth coordination position could therefore be available for other ligands. Treatment with sodium dithionite increased and shifted the Soret band peak toward the maximum at 425–427 nm (Figure 4B and Figure 5). In addition, the maxima of clear α and β bands were detected at 559 nm and 528–529 nm, respectively. Based on these observations, one may conclude that under reducing conditions hexa-coordinate low-spin ferrous heme protein may exist. When the reduced Bvu-heme complex was re-oxidized, maxima at 393 nm and 611 nm were observed (Figure 5A). These features were also visible when difference UV-visible spectra were recorded (Figure 4C,D) and colors of heme-protein samples were examined by visual inspection (Figure 5B).

Under reducing conditions, the plot showing the difference absorbance at 427 nm versus heme concentration demonstrated a Bvu monomer:heme stoichiometry of 1:1 and heme dissociation constant (*K*_d_) = 1.6 ± 0.6 × 10^−9^ M (Figure 4E). Very weak binding of Fe(III)heme did not allow for proper *K*_d_ calculation. All these data suggest that Bvu binds heme preferentially under reducing conditions, similarly to HmuY homologs, mainly to *T. forsythia* Tfo [23], which is in contrast to heme binding by *P. gingivalis* HmuY [23,24,33,34].

Previously, using site-directed mutagenesis, we demonstrated that Met129 and Met158 in PinA and Met119 in PinO could be engaged directly in heme binding [24]. Interestingly, the MD simulations demonstrated that when the heme iron is coordinated by Met119 in PinO, both Met45 and Met145 may interchangeably participate in heme iron coordination. Here, we extended our knowledge to Tfo by providing data corroborating engagement of two Met residues in heme iron coordination in the wild-type protein (Figure 6). Singly substituted Tfo Met149Ala and Met123Ala mutagenesis variants showed changes in the UV-visible spectra, but evident differences were observed in the double Met123Ala/Met149Ala mutagenesis variant, suggesting that Tfo coordinates heme iron using two Met residues.

To study the effects of specific amino acids on heme binding to Bvu, we systematically replaced several Met residues indicated by amino acid sequence comparisons, available crystallographic data, and theoretical modeling by an Ala residue and analyzed the ability of the protein variants to bind heme. UV-visible spectra demonstrated that Met145 and Met172 could be engaged directly in heme iron coordination (Figure 7). This observation was confirmed by significantly diminished heme binding to the double Met145Ala/Met172Ala mutagenesis variant. Similar to HmuY homologs [23,24], Bvu could use two Met residues to coordinate heme iron, which is in contrast to HmuY [33,34], which uses two His residues.

### 2.4. B. vulgatus Bvu May Sequester Heme from Host Serum Albumin but Only under Reducing Conditions

In vivo, free heme is not readily available because it is toxic and is therefore rapidly bound by host heme-scavenging proteins which maintain the concentration of the free heme at very low levels [19]. Our previous studies revealed that the black-pigmented anaerobes *P. gingivalis* and *P. intermedia* display a novel heme acquisition mechanism, whereby oxyhemoglobin is first oxidized to methemoglobin. In the case of *P. gingivalis*, the generation of methemoglobin involves the arginine-specific gingipain protease A (HRgpA) [35], and the cysteine protease interpain A (InpA) in the case of *P. intermedia* [36]. Interestingly, with respect to heme acquisition by black-pigmented anaerobic species, we recently demonstrated that the presence of *Pseudomonas aeruginosa* pyocyanin facilitates extraction of heme from hemoglobin by the *P. gingivalis* HmuY by rapidly oxidizing oxyhemoglobin to methemoglobin [37]. Therefore, our novel paradigm of heme acquisition, which is displayed by the black-pigmented anaerobes, appears to extend to coinfections with other bacteria and offers a mechanism for the ability of *P. gingivalis* to obtain sufficient heme in the host environment. In line with this hypothesis, we demonstrated that a housekeeping protein also exhibiting a moonlighting function, namely GAPDH produced by *S. gordonii*, may serve as a heme donor for *P. gingivalis* [25]. HmuY is also able to compete with albumin, which is the normal frontline heme scavenger in vivo [35], as well as acquire heme from hemopexin [23]. Other studies demonstrated, however, that instead of limiting availability of heme, human serum albumin may promote heme utilization by pathogens [38].

To determine whether the HmuY homolog from *B. vulgatus* can function as a hemophore-like protein, we examined its interaction with albumin. This hemoprotein can be an important heme source for pathogenic bacteria, especially those spreading into different sites of the human body and therefore participating in the onset and progression of systemic inflammatory diseases. As shown in Figure 8 (top panel), Bvu was able to efficiently capture heme which had been bound to human serum albumin, but only under reducing conditions. This effect could be explained by the lower affinity of albumin for Fe(II)heme compared to Fe(III)heme [39,40], which may facilitate heme capture by Bvu. The reducing conditions would also influence the properties of iron coordination by Met residues more effectively as compared to His residues [41]. Such an effect could be explained by the theory of hard and soft acids and bases, demonstrating that Met-ligand binding would be destabilized under oxidizing conditions [42]. In contrast to HmuY homologs characterized so far, although *P. gingivalis* HmuY used two His residues to coordinate heme iron, it was able to efficiently sequester heme from the albumin-heme complex under both air (oxidizing) and reducing conditions (Figure 8, bottom panel).

To analyze possible competition in heme acquisition between *P. gingivalis* and *B. vulgatus*, we examined the interaction occurring between HmuY and Bvu. We demonstrated that Bvu was able to capture heme bound to HmuY under reducing conditions only, but with lower and decreasing ability during incubation time (Figure 9, top panel). Therefore, we hypothesize that under reducing conditions, heme distribution between HmuY and Bvu could exist. In contrast, HmuY efficiently sequestered Fe(III)heme which had been bound to Bvu (Figure 9, bottom panel). Based on these results, one may conclude that heme bound to Bvu might represent a heme reservoir for *P. gingivalis* and serve as its virulence factor. Other studies showed that competition for heme enhanced the pathogenic potential of *P. gingivalis* also during its interaction with *C. albicans* in a mixed biofilm [30].

An oxygen gradient exists not only within the human gastrointestinal tract but also across the intestinal wall [43,44]. The colon niche and colonic mucosa exhibit an anaerobic environment [45], allowing for *P. gingivalis* colonization. Being an anaerobic, asaccharolytic, and highly proteolytic species, *P. gingivalis* may degrade the mucins and extracellular matrix components in the colon, infiltrate the mucus layer, invade the mucosa, and degrade immunological factors which increasing the potential for local and systemic inflammation. Deeper layers of the intestinal wall are rather normoxic [46,47]. We demonstrated that *P. gingivalis* HmuY is structurally stable and can bind heme also under oxidizing conditions, including those caused by the influence of hydrogen peroxide produced by *S. gordonii* or host immune cells [48,49]. Therefore HmuY, regardless of redox conditions, may be beneficial in heme binding compared to Bvu, enabling better growth and virulence of *P. gingivalis*, thus reducing *B. vulgatus* abundance and resulting in dysbiosis in the gut microbiome.

### 2.5. B. vulgatus Bvu Is Less Stable as Compared to P. gingivalis HmuY

One may not exclude the possibility that lower ability of heme binding to Bvu could result from its lower stability, causing heme loss under the conditions studied. Using thermal denaturation analysis, we showed that although the Bvu protein was quite resistant to thermal unfolding, its refolding, similar to HmuY, was unsuccessful (Table 1 and Appendix A). *P. gingivalis* HmuY and *B. vulgatus* Bvu exhibited similar features in both apo form and heme-complexed form, with HmuY being only slightly more resistant to thermal denaturation. Among HmuY homologs examined in this study, only *P. intermedia* PinO showed the highest conformational stability and refolding ability, which confirmed its resistance to thermal denaturation suggested before [24].

Previously, we demonstrated that *P. gingivalis* HmuY is completely resistant to proteolytic activity performed by a variety of proteases, including *P. gingivalis* gingipains and enzymes produced by other periodontopathogens, as well as enzymes produced by host immune cells, which is in contrast to its homologs [23,24,33,36,37]. It is also worth noting that HmuY not only can be found as a surface-associated lipoprotein, but it also can be spread into the environment in a form integrated with outer-membrane vesicles, as well as in the form of soluble protein shed from the bacterial cell surface [23,24,29,33]. Such features might be advantageous in HmuY spreading in the different host niches and therefore increasing heme acquisition ability by *P. gingivalis*. Compared to HmuY, Bvu was degraded during *P. gingivalis* growth (Figure 10). However, neither HmuY nor Bvu was digested when added to *B. vulgatus* cultures. These data suggest that high proteolytic activity of *P. gingivalis* and HmuY resistance to proteases may be advantageous for this pathogen’s survival also in the gut microbiome.

### 2.6. Conclusions

This is the first report demonstrating that *B. vulgatus* expresses a heme-binding protein with properties of hemophore-like proteins, thus increasing the number of members of the novel HmuY-like family of proteins. Our analysis confirms that various members of this family may have developed specific heme-binding pockets and the ability to sequester heme from host hemoproteins or even to acquire heme bound to heme-binding proteins produced by cohabitating bacteria. The independent evolution of these properties has resulted in different mechanisms of heme coordination in HmuY (two His residues) versus Bvu (two Met residues). The former binds heme very effectively both in oxidizing and reducing environments, while Bvu appears to prefer reducing conditions. Moreover, our findings demonstrate that the *P. gingivalis* Hmu heme acquisition system, with the leading role played by HmuY, may compete with other members of HmuY-like proteins to increase *P. gingivalis* virulence and its ability to cause dysbiosis also in the gut microbiome.

Data gained in this study further confirm the importance of HmuY in the context of *P. gingivalis* survival in regard to its ability to cause dysbiosis not only in the oral cavity but also in the gut microbiome. In a wider context, clarification of these mechanisms is important because of the need to characterize one of the basic mechanisms that allows the survival of bacteria in the hostile environment of the host and that plays a key role in dysbiosis, especially during the course of infections accompanying inflammatory systemic diseases. Moreover, the data reported here might offer insights into the development of strategies to reduce the potential of heme availability as a means to control *P. gingivalis* growth and virulence in vivo.

## 3. Materials and Methods

### 3.1. Bacterial Stains and Growth Conditions

*B. vulgatus* ATCC 8482 (Pol-Aura, Olsztyn, Poland) was grown anaerobically at 37 °C for 3 days on Brain-Heart Infusion (BHI) agar (Biomaxima, Lublin, Poland) plates supplemented with 7.7 μM heme (Fluka, Munich, Germany), 0.05% L-cysteine (Sigma, St. Louis, MO, USA) and 0.2% NaHCO_3_ (Fluka). These cultures were inoculated into liquid Brain-Heart Infusion medium (BHI; Biomaxima) supplemented with 0.05% L-cysteine and 0.2% NaHCO_3_. To grow bacteria under high-iron/heme conditions (Hm), this medium was supplemented with 7.7 μM heme (Fluka). To grow bacteria under low-iron/heme conditions (DIP), heme was not added, and iron was chelated by addition of 160 μM 2,2-dipyridyl. The *P. gingivalis* A7436 strain was grown under the conditions described for *B. vulgatus* on blood agar plates (ABA; Biomaxima). Cultures were inoculated into liquid basal medium (BM) prepared from 3% trypticase soy broth (Becton Dickinson), 0.5% yeast extract (Biomaxima), 0.5 mg/l menadione (Fluka), 0.05% L-cysteine, and 7.7 μM heme. The *Escherichia coli* Rosetta (DE3)RIL strain (Agilent Technologies, Santa Clara, CA, USA) was cultured under aerobic conditions.

### 3.2. Plasmid Construction, Site-Directed Mutagenesis, Overexpression, and Purification of Proteins

The gene-encoding Bvu protein was amplified using PCR (primers are listed in Table 2) and inserted into the pMAL-c5x_His plasmid as described previously [50]. *B. vulgatus* Bvu protein (GenBank ID: ABR39853), lacking the predicted signal peptide sequence (MKRYLSIITILGMMLLPFSAC), was overexpressed in *E. coli* Rosetta (DE3)RIL cells. The *P. gingivalis* HmuY, *T. forsythia* Tfo, *P. intermedia* PinO and PinA, and *B. vulgatus* Bvu proteins were purified from a soluble fraction obtained from *E. coli* cell lysate as described previously [23,24].

Concentration of the purified proteins was determined spectrophotometrically using the empirical molar absorption coefficients [23,24,33]. In this study, the empirical molar absorption coefficient determined for Bvu was 39519 M^−1^cm^−1^.

For sodium dodecyl sulfate-polyacrylamide gel electrophoresis (SDS-PAGE), samples were prepared and analyzed as reported previously [23,24].

Point mutations were introduced into the gene-encoding Bvu or Tfo protein using a QuikChange II XL Site-Directed Mutagenesis Kit (Agilent Technologies). Selected amino acids, based on amino acid alignment [51] and the comparison of 3-dimensional protein structures visualized using the program Chimera UCFS [52], with a potential ability to coordinate heme iron, were substituted by an alanine, resulting in single- or double-point mutations. Site-directed mutagenesis primers are available upon request.

### 3.3. Heme-Protein Complex Formation

Heme (hemin chloride; ICN Biomedicals, Aurora, OH, USA) solutions were prepared as reported previously [23,35]. Formation of heme-protein complexes was examined in 20 mM sodium phosphate buffer, pH 7.4, containing 140 mM NaCl (PBS). UV-visible spectra were recorded in the range 250–700 nm with a double-beam Jasco V-650 spectrophotometer using cuvettes with 10 mm path length. Titration curves were analyzed using the equation for a 1-site binding model, and dissociation constant (*K*_d_) values determined as reported earlier [23,24,53] using OriginPro 8 software (OriginPro Corporation, Northampton, MA, USA). To analyze the redox properties of the heme iron, 1 mM or 10 mM sodium dithionite prepared in PBS was used as the reductant, and 1 mM potassium ferricyanide prepared in PBS as the oxidant [23,24,34].

### 3.4. Heme Sequestration Experiments

Protein-heme complexes were prepared by incubating a 120 μM stock solution of human albumin (Sigma; A-8763), Bvu or HmuY in PBS at 37 °C with heme at a 1:1.2 protein-to-heme molar ratio and subsequently passed through Zeba Spin Desalting columns (Thermo Fisher, Scientific, Waltham, MA, USA) to ensure that no uncomplexed heme remained. Coincubation of Bvu with HmuY or with human serum albumin was carried out in PBS at 37 °C and monitored by UV-visible spectroscopy under air (oxidizing) or reducing conditions using each protein at 5 µM concentration [23,24,35].

### 3.5. Determination of Protein Stability

To determine conformational stability of the purified proteins, thermal unfolding and refolding were examined. Proteins in apo form or in a complex with heme were prepared in PBS and examined using a label-free fluorimetric analysis with Prometheus NT.48 apparatus (Nano Temper Technologies, Munich, Germany). NanoDSF grade capillaries were filled with 1 mg/mL protein solutions and heated from 25 °C to 90 °C with a 1 °C/min heating rate at low detector sensitivity with an excitation power of 8%. Unfolding and refolding transition points (T_m_) were determined from the first derivative of the changes in the emission wavelengths of tryptophan and tyrosine fluorescence at 330 nm and 350 nm, respectively, automatically identified by the Prometheus NT.48 control software (Nano Temper Technologies). Two independent measurements were performed. The statistical analysis was performed using Student’s *t*-test. Data were expressed as mean ± SD. For statistical analysis, the GraphPad software (GraphPad Prism 5.0 Inc., San Diego, CA, USA) was used.

### 3.6. Proteolytic Digestion

To examine the susceptibility of the Bvu to proteases, *P. gingivalis* and *B. vulgatus* cells were grown under rich, high-iron/heme conditions (ensuring proper cell viability and efficient proteolytic activity) in the presence of added purified HmuY and Bvu proteins at a final concentration of 5 μM [23,24]. All cultures (1 mL) were started at OD_600_ = 0.2 for *P. gingivalis* and OD_600_ = 0.4 for *B. vulgatus*, grown and collected at 0 h, 6 h, and 24 h. The number of bacterial cells at the starting point was ~2 × 10^8^ or ~4 × 10^8^ per ml of the culture medium for *P. gingivalis* and *B. vulgatus*, respectively, and increasing during cultivation. As controls, *P. gingivalis* or *B. vulgatus* cultures without addition of the purified proteins or culture medium alone were analyzed. At the indicated timepoints, aliquots of samples were examined by SDS-PAGE and Coomassie Brilliant Blue G-250 (CBB) staining.

### 3.7. Quantitative Reverse Transcriptase-Polymerase Chain Reaction (qRT-PCR)

RNA was purified from 0.5 × 10^8^ − 4 × 10^8^
*B. vulgatus* cells as described previously [23,24]. Reverse transcription was carried out with 200 ng of RNA using a LunaScript RT SuperMix Kit (New England Biolabs). PCR was performed using a SensiFAST SYBR No-ROX Kit (Bioline, London, UK) and the LightCycler 96 System (Roche, Basel, Switzerland). The amplification reaction was carried out as follows: An initial denaturation at 95 °C for 2 min, 40 cycles of denaturation at 95 °C for 5 s, primer annealing at 60 °C for 10 s, and extension at 72 °C for 20 s. The melting curves were analyzed to monitor the quality of PCR products. Relative quantification of the *bvu* gene was determined in comparison to the *16S rRNA* gene of *B. vulgatus* (GenBank ID: AJ867050) as a reference using the ΔΔC_t_ method. All samples and controls were run in triplicate in 3 independent experiments for the target and reference genes. All primers are listed in Table 2. The statistical analysis was performed using Student’s *t*-test. Data were expressed as mean ± SD. For statistical analysis, the GraphPad software (GraphPad Prism 5.0 Inc.) was used.

## Figures and Tables

**Figure 1 ijms-22-02237-f001:**
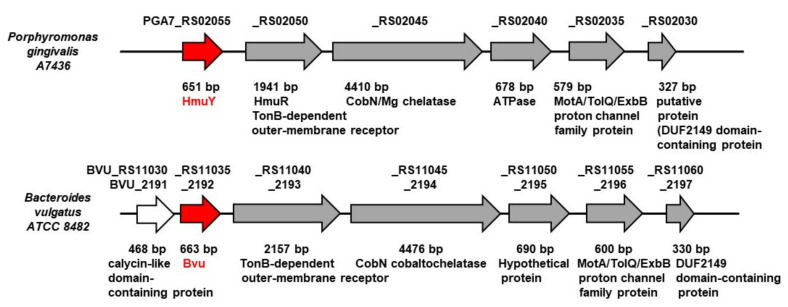
Organization of *P. gingivalis hmu* operon and *hmu*-like operon in *B. vulgatus.* Red arrows indicate genes encoding HmuY and Bvu proteins.

**Figure 2 ijms-22-02237-f002:**
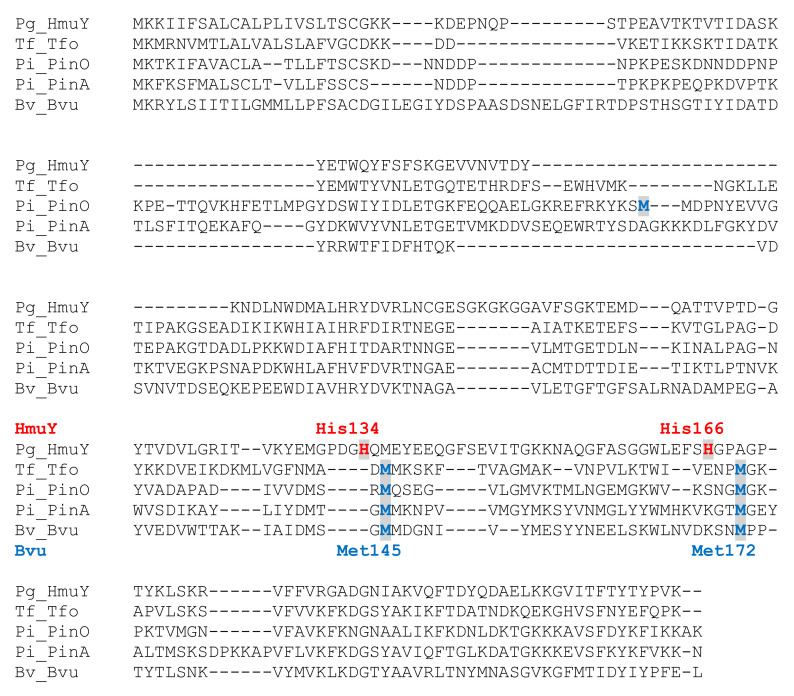
Amino acid sequence comparison of Bvu protein with the best characterized HmuY family members. Verified by mutagenesis and crystallization (*P. gingivalis* HmuY) or examined by mutagenesis (*T. forsythia* Tfo, *P. intermedia* PinO and PinA, *B. vulgatus* Bvu) heme iron-coordinating ligands are shown in grey boxes and marked in red (HmuY) or blue (HmuY homologs).

**Figure 3 ijms-22-02237-f003:**
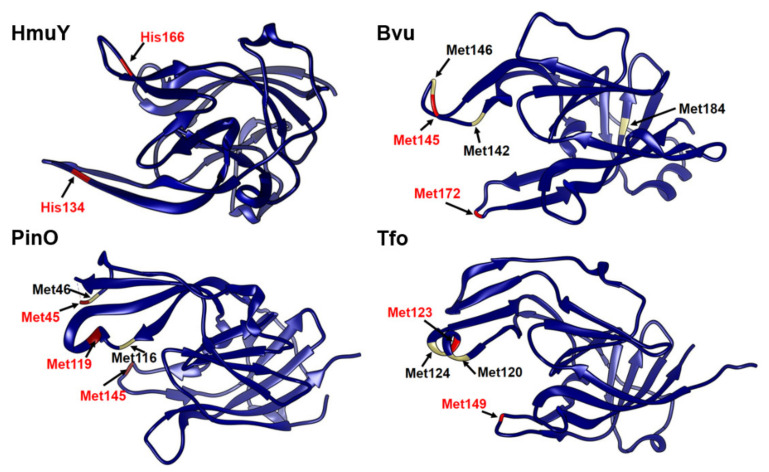
Three-dimensional protein structures of HmuY and its homologs. Apo-protein structures of *P. gingivalis* HmuY (PDB ID: 6EWM), *P. intermedia* PinO (PDB ID: 6R2H), *T. forsythia* Tfo (PDB ID: 6EU8), and *B. vulgatus* Bvu (PDB ID: 3U22) are shown. Experimentally confirmed amino acids coordinating heme iron are shown in red. Other putative amino acids potentially involved in heme iron binding in HmuY homologs, examined by site-directed mutagenesis, are shown in black. Protein structures were visualized using the program Chimera UCFS.

**Figure 4 ijms-22-02237-f004:**
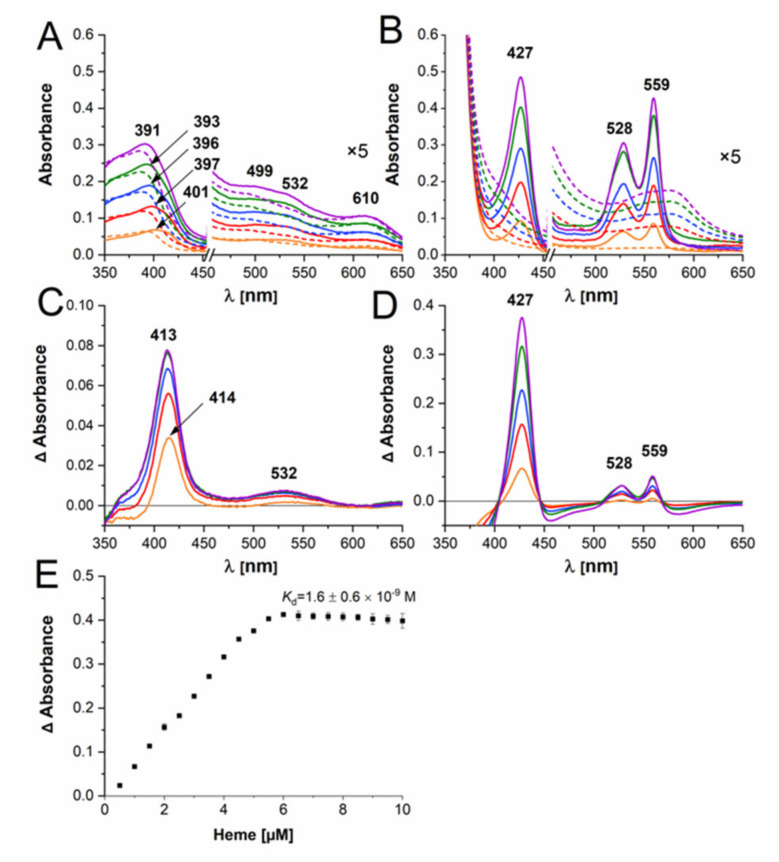
Spectroscopic analysis of heme binding to the purified Bvu protein. In the analysis, 5 μM protein was titrated with heme under air (oxidizing) conditions (**A**) and subsequently reduced by sodium dithionite (**B**). Titration of Bvu with heme (solid lines) and buffer with heme (dashed lines) is shown (**A**,**B**). Spectra showing increasing concentration of heme or protein-heme complex are shown with different colors. Difference spectra examined under air (oxidizing) (**C**) and reducing conditions (**D**) are also shown. Spectra shown with different colors demonstrate increasing concentration of protein-heme complex after subtraction of heme-only spectra. (**E**) The curve was generated by titration of 5 μM protein with heme by measuring the difference spectra between the protein + heme and heme-only samples under reducing conditions. Results are shown as mean ± SD from three independent experiments.

**Figure 5 ijms-22-02237-f005:**
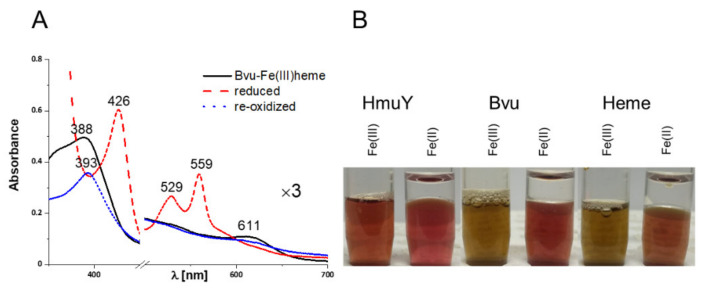
Spectroscopic analysis of heme binding to the purified Bvu protein. (**A**) In the analysis, 5 μM protein in complex with heme was monitored under air (oxidizing conditions; black solid line), subsequently reduced by sodium dithionite (red dashed line), and re-oxidized by potassium ferricyanide (blue dotted line). (**B**) Colors of 100 μM proteins complexed with heme (protein:heme molar ratio 1:1) in 20 mM sodium phosphate buffer, pH 7.4, containing 140 mM NaCl (PBS).

**Figure 6 ijms-22-02237-f006:**
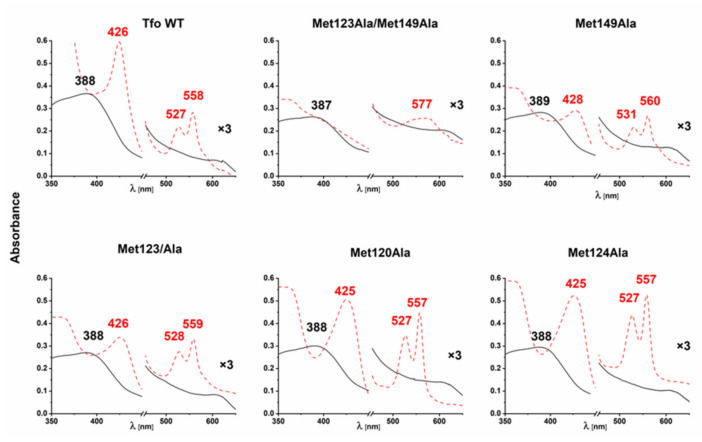
UV-visible absorption analysis of *T. forsythia* Tfo site-directed mutagenesis variants complexed with heme. Spectra were recorded for the protein-Fe(III)heme complexes (solid, black line) and protein-Fe(II)heme complexes (dashed, red line). All spectra were recorded at a 1:1 protein:heme molar ratio for singly or dually replaced methionine residues by alanine residues. WT, wild-type protein.

**Figure 7 ijms-22-02237-f007:**
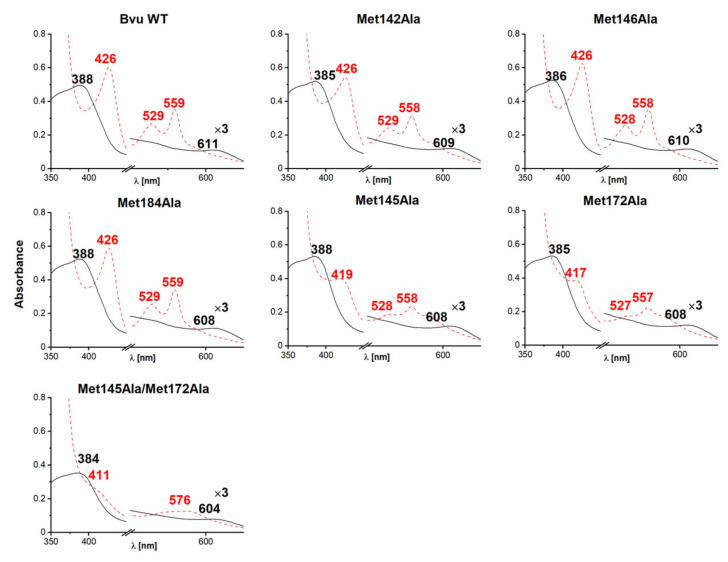
UV-visible absorption analysis of *B. vulgatus* Bvu site-directed mutagenesis variants complexed with heme. Spectra were recorded for the protein-Fe(III)heme complexes (solid, black line) and protein-Fe(II)heme complexes (dashed, red line). All spectra were recorded at a 1:1 protein:heme molar ratio for singly or dually replaced methionine residues by alanine residues. WT, wild-type protein.

**Figure 8 ijms-22-02237-f008:**
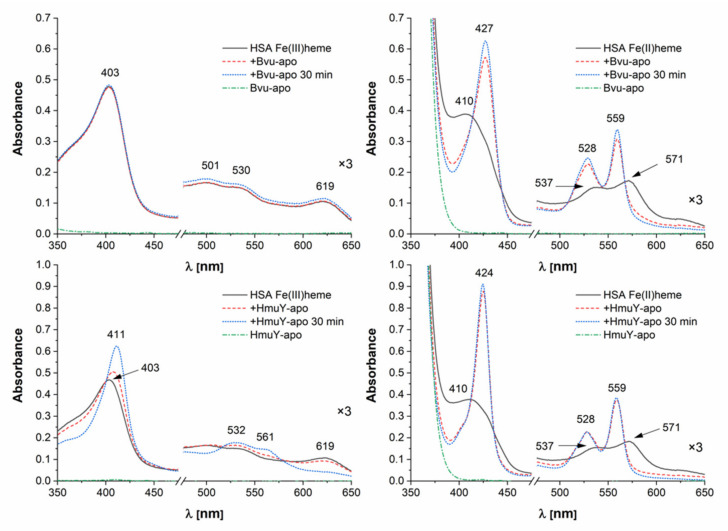
Heme sequestration from host hemoprotein. *B. vulgatus* Bvu or *P. gingivalis* HmuY proteins were incubated with an equimolar concentration of human serum albumin (HSA) under air (oxidizing) conditions (left panel) or reducing conditions formed by addition of sodium dithionite (right panel). Changes in absorption spectra analyzed by UV-visible spectroscopy are shown at indicated timepoints.

**Figure 9 ijms-22-02237-f009:**
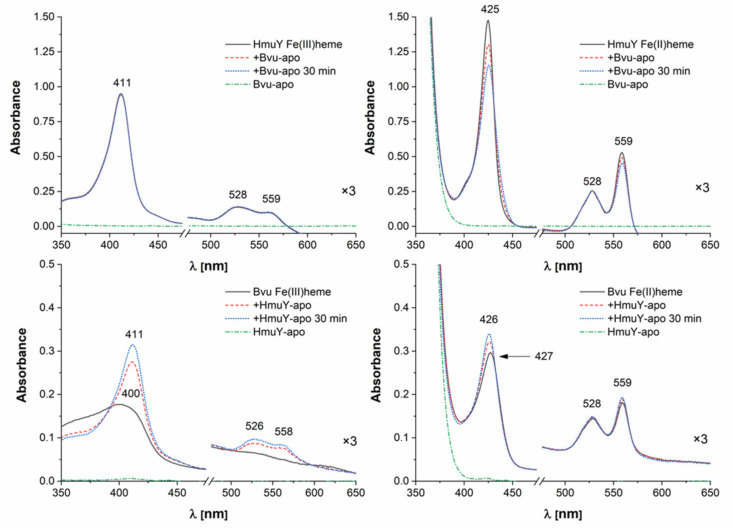
Heme sequestration from *P. gingivalis* HmuY. *B. vulgatus* Bvu protein was incubated with equimolar concentration of HmuY under air (oxidizing) conditions (left panel) or reducing conditions formed by addition of sodium dithionite (right panel). Changes in absorption spectra analyzed by UV-visible spectroscopy are shown at indicated timepoints.

**Figure 10 ijms-22-02237-f010:**
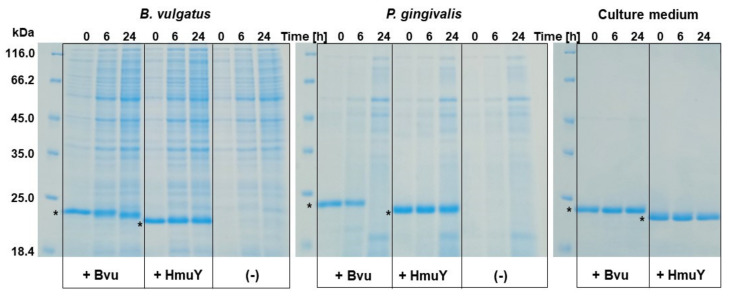
Susceptibility of *B. vulgatus* Bvu protein to bacterial proteases. Proteolytic digestion was examined by growing bacteria under high-iron/heme conditions in the presence of purified HmuY or Bvu proteins (marked with asterisks). Samples collected at indicated time points were separated by SDS-PAGE and proteins stained with CBB.

**Table 1 ijms-22-02237-t001:** Analysis of conformational stability of hemophore-like proteins. Thermal unfolding and refolding transition points (T_m_) were determined from the first derivative of the changes in the emission wavelengths of tryptophan and tyrosine fluorescence at 330 nm and 350 nm using a label-free fluorometric analysis. Two independent measurements were performed. ND, not detected.

Protein	Unfolding (°C)		Refolding (°C)	
	Apo-Protein	Protein-Heme	Apo-Protein	Protein-Heme
*P. gingivalis* HmuY	70.8 ± 0.1	70.6 ± 0.2	ND	ND
*T. forsythia* Tfo	61.4 ± 0.2	61.2 ± 0.3	ND	ND
*P. intermedia* PinO	69.3 ± 0.2	68.8 ± 0.3	63.9 ±0.2	63.8 ± 0.3
*P. intermedia* PinA	62.3 ± 0.3	62.0 ± 0.2	ND	ND
*B. vulgatus* Bvu	60.7 ± 0.1	60.4 ± 0.1	ND	ND

**Table 2 ijms-22-02237-t002:** Primers designed and used in this study.

Name	DNA Sequence (5′→3′)	Description
16S_F	ACACGTATCCAACCTGCCGT	Amplify fragment of *B. vulgatus 16S rRNA* gene used in qRT-PCR
16S_R	ATGGAACGCATCCCCATCGT	
Bvu_F	AGACCCTTCTACCCATTCGGG	Amplify fragment of *B. vulgatus bvu* gene used in qRT-PCR
Bvu_R	CCCATTCCTCCGGTTCTTTCTG	
Bvu_FP	CAACCTCGGGATCGAGGGAAGGATGGATGGTATTTTGGAAGGAATA	Amplify *bvu* gene for overexpression and purification of recombinant N-terminally tagged Bvu protein with 6His tag and maltose-binding protein (MBP), lacking predicted signal peptide sequence
Bvu_RP	TATTTAATTACCTGCAGGGAATTCGGATCCTTACAATTCAAACGGATAAATATAATC	

## Data Availability

The data obtained in this study are presented in the paper, in Appendix A, or available upon request from the corresponding author.

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
