# Peer review of "Porphyromonas gingivalis HmuY and Bacteroides vulgatus Bvu—A Novel Competitive Heme Acquisition Strategy"

_ijms, 2021, doi:10.3390/ijms22052237_

Round 1
Reviewer 1 Report
The manuscript by Klaudia Siemińska et al. illustrated that the P. gingivalis Hmu heme acquisition system, with the leading role played by HmuY, may compete with other members of HmuY-like proteins, such as Bvu, to increase P. gingivalis virulence. The study is interesting and informative. It provides a new model to explain the virulence of P. gingivalis.
I have two questions. First, are Bvu-like or HmuY-like proteins very common in the oral microbiome? Could the authors provide more data on this question? I suspect they would be very common in the oral microbiome.
Second, are there other proteins that P. gingivalis or B. vulgatus could use to get heme or iron? This should be discussed in the manuscript.
Author Response
I have two questions. First, are Bvu-like or HmuY-like proteins very common in the oral microbiome? Could the authors provide more data on this question? I suspect they would be very common in the oral microbiome.
Additional explanation regarding HmuY expression was added in the Results and Discussion section (page 2, lines 90-96 and page 9, lines 284-287) of the revised manuscript.
“Previously, we demonstrated that P. gingivalis HmuY is constitutively expressed, but significantly higher levels of HmuY at both mRNA and protein levels, when the bacterium grew in vitro under low-iron/heme conditions or as a biofilm constituent, environment typical for healthy oral cavity or early stages of chronic periodontitis, can be observed [31,32]. We also demonstrated that HmuY is important for bacterial survival and invasion of human cells [20]. Significantly higher HmuY expression in P. gingivalis grown in the form of biofilm together with Candida albicans [23] and in patients with periodontitis [24,25] confirmed its requirement under in vivo conditions.”
“It is also worth noting that HmuY not only can be found as a surface-associated lipoprotein, but also can be spread into the environment in a form integrated with outer-membrane vesicles, as well as in the form of soluble protein shed from the bacterial cell surface [26,27,32,33].”
Analysis of Bvu expression was not previously reported and will be shown for the first time in our manuscript.
Second, are there other proteins that P. gingivalis or B. vulgatus could use to get heme or iron? This should be discussed in the manuscript.
We agree with the Reviewer that information provided in the Introduction section was not sufficient. Therefore, other iron/heme uptake systems were mentioned in the Introduction section (page 2, lines 50-56 and page 2, lines 61-62) of the revised manuscript.
“P. gingivalis does not produce and use siderophores to gain iron, lacks the ability to synthesize protoporphyrin IX (PPIX), and therefore must utilize mechanisms to acquire these compounds. Among them is the best characterized Hmu system composed of a hemophore-like, heme-binding protein (HmuY), TonB-dependent outer-membrane receptor involved in heme transport through the outer membrane (HmuR), and four uncharacterized proteins with unknown functions [19]. Other best-characterized P. gingivalis mechanisms of iron and heme uptake comprise Hus system and a set of cysteine proteases, gingipains [19].”
“To date, iron and heme acquisition systems in B. vulgatus, dominant member of the healthy gut microbiome, have been not characterized.”
Reviewer 2 Report
I have read “Porphyromonas gingivalis HmuY and Bacteroides vulgatus Bvu – novel competitive heme acquisition strategy in the gut microbiome". In this manuscript, authors investigated novel heme acquisition protein, Bvu, and compared Bvu and other heme acquisition proteins. binding of heme and Bvu. Heme binding property of Bvu is informative. I have just some concerning point in this manuscript.
- The title is not adequate for this study.
Authors did not study the gut microbiota, the word “the gut microbiota” in the title should be deleted.
- Figure 4A-D, please add information of each spectrum lines.
Author Response
- The title is not adequate for this study.
Authors did not study the gut microbiota, the word “the gut microbiota” in the title should be deleted.
We agree with the Reviewer and this statement was removed from the title. In the revised manuscript, only possibility of an interaction in the gut microbiome between HmuY produced by oral bacterium (P. gingivalis), which is able to invade (we removed the strong, not confirmed statement “colonize” too) the gut, and between Bvu produced by a dominant gut bacterium (B. vulgatus) is discussed.
- Figure 4A-D, please add information of each spectrum lines.
Lines showing spectra in the Figure 4A-D are now explained with more details in the legend to this figure (page 5).
“Figure 4. Spectroscopic analysis of heme binding to the purified Bvu protein. 5 μM protein was titrated with heme under air (oxidizing) conditions (A) and subsequently reduced by sodium dithionite (B). Titration of Bvu with heme (solid lines) and buffer with heme (dashed lines) is shown (A, B). Spectra showing increasing concentration of heme or protein-heme complex are shown with different colors. Difference spectra examined under air (oxidizing) (C) and reducing conditions (D) are also shown. Spectra shown with different colors demonstrate increasing concentration of protein-heme complex after subtraction of heme-only spectra. (E) The curve was generated by titration of 5 μM protein with heme by measuring the difference spectra between the protein+heme and heme-only samples under reducing conditions. Results are shown as mean±SD from three independent experiments.”
Reviewer 3 Report
First of all, it should be noted that the paper presented to me for review (“Porphyromonas gingivalis HmuY and Bacteroides vulgatus Bvu – novel competitive heme acquisition strategy in the gut microbiome”) deals with important and current problems of gastrointestinal microbiota.
The authors showed good knowledge of the research topic. They applied appropriate and modern research methodology to achieve the intended purpose of the research. The results of the study are presented in 10 figures and 2 tables in a clear and transparent way.
In classical scientific papers, the methodology section should come before the results and discussion. Therefore, I suggest the authors consider changing the order of the chapters.
I also recommend including a few sentences of conclusions at the end of the article summarizing the obtained research results.
I have no major comments on the publication. I propose to accept the work for publication after minor corrections.
Author Response
Reviewer #3
First of all, it should be noted that the paper presented to me for review (“Porphyromonas gingivalis HmuY and Bacteroides vulgatus Bvu – novel competitive heme acquisition strategy in the gut microbiome”) deals with important and current problems of gastrointestinal microbiota.
The authors showed good knowledge of the research topic. They applied appropriate and modern research methodology to achieve the intended purpose of the research. The results of the study are presented in 10 figures and 2 tables in a clear and transparent way.
In classical scientific papers, the methodology section should come before the results and discussion. Therefore, I suggest the authors consider changing the order of the chapters.
We used the journal’s template to prepare the manuscript, where Materials and Methods section is placed after the Results and Discussion section.
I also recommend including a few sentences of conclusions at the end of the article summarizing the obtained research results.
We included additional details to our summary with subsequent conclusions in the 2.6. Conclusion paragraph (former name of this paragraph was 2.6. Final remarks) (page 10, lines 307-326).
I have no major comments on the publication. I propose to accept the work for publication after minor corrections.